# Deep Learning to Measure the Intensity of Indocyanine Green in Endometriosis Surgeries with Intestinal Resection

**DOI:** 10.3390/jpm12060982

**Published:** 2022-06-16

**Authors:** Alicia Hernández, Pablo Robles de Zulueta, Emanuela Spagnolo, Cristina Soguero, Ignacio Cristobal, Isabel Pascual, Ana López, David Ramiro-Cortijo

**Affiliations:** 1Department of Obstetrics and Gynecology, Hospital Universitario La Paz, Paseo de la Castellana, 261, 28046 Madrid, Spain; aliciahernandezg@gmail.com (A.H.); ignaciocristobal94@gmail.com (I.C.); analopezcarrasco.lopez@gmail.com (A.L.); 2Department of Obstetrics and Gynecology, Faculty of Medicine, Universidad Autónoma de Madrid, C/Arzobispo Morcillo 2, 28029 Madrid, Spain; 3Department of Signal Theory and Communications, Telematics and Computing Systems, Universidad Rey Juan Carlos, Camino del Molino, 5, D201, Departamental III, 28942 Fuenlabrada, Spain; pabloroblesdz@gmail.com; 4Department of General Surgery, Hospital Universitario La Paz, Paseo de la Castellana, 261, 28046 Madrid, Spain; isabelpasmi@hotmail.com; 5Department of Physiology, Faculty of Medicine, Universidad Autónoma de Madrid, C/Arzobispo Morcillo 2, 28049 Madrid, Spain; david.ramiro@uam.es

**Keywords:** deep endometriosis, deep learning, video protocol, automatic segmentation, bowel resection, laparoscopy

## Abstract

Endometriosis is a gynecological pathology that affects between 6 and 15% of women of childbearing age. One of the manifestations is intestinal deep infiltrating endometriosis. This condition may force patients to resort to surgical treatment, often ending in resection. The level of blood perfusion at the anastomosis is crucial for its outcome, for this reason, indocyanine green (ICG), a fluorochrome that green stains the structures where it is present, is injected during surgery. This study proposes a novel method based on deep learning algorithms for quantifying the level of blood perfusion in anastomosis. Firstly, with a deep learning algorithm based on the U-Net, models capable of automatically segmenting the intestine from the surgical videos were generated. Secondly, blood perfusion level, from the already segmented video frames, was quantified. The frames were characterized using textures, precisely nine first- and second-order statistics, and then two experiments were carried out. In the first experiment, the differences in the perfusion between the two-anastomosis parts were determined, and in the second, it was verified that the ICG variation could be captured through the textures. The best model when segmenting has an accuracy of 0.92 and a dice coefficient of 0.96. It is concluded that segmentation of the bowel using the U-Net was successful, and the textures are appropriate descriptors for characterization of the blood perfusion in the images where ICG is present. This might help to predict whether postoperative complications will occur during surgery, enabling clinicians to act on this information.

## 1. Introduction

Endometriosis is a chronic benign gynecological disease that involves the implantation and growth of endometrial tissue outside the endometrium, including extrauterine locations [1]. It is estimated that 6–15% of childbearing age women suffer endometriosis [1,2]. Among those women, the prevalence oscillates between 30 and 50%, if they present chronic pelvic pain or infertility [2]. The intestinal deep endometriosis (DE), the most severe category of endometriosis [3], affects up to 37% of women with endometriosis [4,5,6]. Currently, surgery is the only solution when medical treatment is insufficient to manage pain and disease progression [7]. In addition, it is the therapy that yields the superior cost–benefit and produces an impact on quality of life [7,8]. The surgery to remove DE infiltrating nodules of the bowel is highly recommended when the lesions affect the intestinal, urinary, sexual, and reproductive functions [6]. Procedures employed for nodules removal in DE bowel surgery include shaving, discoid excision, and segmental resection [3,6,9]. The main postoperative difference between the three techniques lies in the rate of complications and recurrence of nodules [3,9,10].

When removing the nodules, some of the surrounding tissue is also eliminated, leading to the adjacent microvasculature [6]. In most cases, the removed arteries were exclusively supplying the damaged portion of the bowel; nevertheless, perfusion of the healthy bowel can be likewise compromised [11]. Therefore, the decrease in blood supply to the healthy bowel might evolve into a postoperative complication [10,12,13]. Thus, the need arises to strike a balance between removing as much extrauterine endometrial tissue as possible and reducing the number of complications [6].

Within conservative surgery, the procedures used for nodules removal are laparotomy and laparoscopy, the latter being the gold standard [1]. Laparoscopy allows for complete lesions excision, less fertility impairment, and better long-term outcomes [1,7,8,14]. The use of laparoscopy in bowel resection surgery, together with fluorescence-guided surgery (FGS) and indocyanine green (ICG), has demonstrated a relationship between the level of perfusion and postoperative complications, decreasing oxygenation effects on the bowel tissue [15]. FGS, which allows the real-time visualization of anatomical structures, could help surgeons to evaluate reperfusion of bowel anastomosis [16]. Moreover, FGS has been proven to enhance surgical outcomes since the election of bowel shaving, discoid excision or segmentary resection can be discussed during the surgery [12,17]. There are systems that accomplish satisfactory results when monitoring the level of perfusion quantitatively [18]. The two most widespread models are fluorescence-based enhanced reality (FLER) [19] and quantification of fluorescence angiography (Q-ICG) [20]. Both fall into the category of software-based quantification algorithms [18] and analyze the level of ICG by processing the fluorescent signal to produce a fluorescence–time curve (FTC) [19,20]. After recording for a fixed period, based on the FTC’s final slope, the quantitative perfusion assessment is performed [18,19,20]. Even if these systems are used, they cannot be considered a fully reliable technique to monitor the level of perfusion quantitatively [17] due to the qualitative approach based on visual judgment being the current gold standard [21,22].

However, to obtain valid quantitatively and comparable measurements of the ICG, it would be necessary to design an imaging protocol to standardize video recordings in gynecological surgeries with bowel resection. Nevertheless, it should be considered that to improve the user experience for surgeons when viewing ICG-colored structures, the laparoscopy towers are equipped with digital filters that attenuate or intensify the green level depending on the distance between the lens and the colored structure [23]. In addition, camera gain can be manually modified by adjusting contrast and brightness. These factors affect how the green is viewed and recorded; requiring the design of a protocol to ensure that the images captured in all surgeries are as similar as possible. Furthermore, there is another existing problem that can make measurements unreliable. As shown by Son et al. [17], once the ICG reaches and completely stains the intestine, the level of intensity captured by the camera increases until it hits a maximum and, from that point, it begins to decrease until it stabilizes.

For all the above-mentioned, the aims of this article were: (1) to find descriptors that quantitatively measure the ICG intensity in videos from the FGSs involving bowel resection, (2) to implement a segmentation algorithm, based on deep learning (DL), that allows automatic division of the bowel, and (3) to generate a standardized and reproducible protocol for video recording of deep bowel endometriosis FGSs with ICG. Additionally, we consider processing a database with the frames extracted from videos of deep endometriosis surgeries.

## 2. Materials and Methods

### 2.1. Pilot Protocol for Video Recording

The proposed protocol followed in this study combines the former manner of capturing the videos with guidelines to ensure the correct standardization of image acquisition. The protocol has three stages and was designed to be used in surgeries involving segmental resection and/or discoid excision:**Optics placement**: The camera needs to be optimally placed before injecting the ICG and just after performing the resection. For this, the lens is directed to the area of the anastomosis and positioned forming a 90° with the anatomical plane (the intestinal serosa). Then, it is positioned in such a way that 2/3 of the proximal part and 1/3 of the distal part are shown. The distance at which the optic should be placed is approximately 8 cm. This distance would correspond to two open fenestrated grasping forceps.**Video recording**: After correctly positioning the optics, 5 mL of ICG is injected (i.v. 2.5 mg/mL). The surgeon should record the same shot for at least 140 s to ensure that the ICG has reached the bowel and is correctly stabilized [17].**Completion**: After successfully recording both areas of the bowel the surgery can be resumed.

The Appendix A is shown an example of what video was considered as appropriate recorded (Appendix A) for the proposed acquisition of frames in this study.

### 2.2. Data Collection and Processing

A database composed of 7 videos of gynecological surgeries recorded in the gynecology surgery rooms from the Hospital Universitario La Paz (HULP, Madrid, Spain) between 2018 and 2020 was assembled for attaining the objectives of this project. All recordings were collected with a Stryker^®^ (Stryker Iberia S.L., Madrid, Spain) or Olympus^®^ (Olympus Iberia S.A.U., Barcelona, Spain) laparoscopy tower during nodule removal interventions. Discoid excision or segmental resection was applied to remove deep infiltrating nodules from the bowel. Given that the videos cover the totality of the surgery (containing scenes that are not of interest to this study), certain extracts were selected. From these, frames were retrieved for posterior analysis. The resolution of the videos was full HD, with each frame having dimensions of 1920 × 1080 × 3. The frames were divided into three categories:Type I (Figure 1A): the intestinal tract, primarily the colon, in healthy conditions or with deep infiltrating nodules.Type II (Figure 1B): the bowel after segmental resection or discoid excision. These scenes were captured with or without ICG, the first ones being used to analyze blood perfusion.Type III (Figure 1C): the proximal part of the bowel outside the peritoneal cavity after a segmental resection. They appear to a lesser extent than the previous ones.

To achieve the goal of automatic segmentation, 651 frames were subtracted from 5 of the videos: 153 were type I, 385 were type II and 113 were type III, reserving 592 for the training process and 59 for testing. Once the frames were selected, binary masks of four different regions of interest (ROI) were produced as follows. The four possible ROIs were the bowel, the proximal and distal part of the bowel with respect to the anastomosis, and the rest of the anatomical plane. For the preparation of these binary masks, firstly the ROIs were manually segmented (directly from the RGB frames) by a trained member of the research team, using the open-source editing tool GIMP (ver. 2.10.30 2021, GPLv3+, Microsoft Windows). Subsequently, the RGB ROIs were converted to greyscale, and, by thresholding and the use of morphological operators [24], specifically two consecutive closings [25] using a 7 × 5 ellipse as structuring element, the binary masks were finally obtained. The image processing tools from the Python library OpenCV (ver. 4.5.5.64) were used. In type I and III categories, the ROI to be segmented was the intestine. In contrast, type II contained 3 ROIs: the intestine and the proximal and distal part of the anastomosis.

Once the binary masks were created, the multiclass masks, with which the deep learning algorithm was trained and evaluated, were generated. Depending on the scene to which the frame corresponded, two different strategies were designed to arrange the multiclass masks. For type II frames, a total of four differentiated areas were established: the background, the proximal part, the distal part, and the intestine. For the type I and III frames, creating the multiclass mask consisted merely of assigning the class background and intestine (Figure 2).

### 2.3. Automatic Segmentation

To adequately assess the level of blood perfusion after surgery, it is necessary to isolate the intestine from the rest of the anatomical plane and, when there is an anastomosis, to distinguish the proximal and distal parts. This helps the surgeons since it ensures that only the ICG present in the bowel is considered when quantifying the level of perfusion. To this end, an automatic segmentation algorithm was employed. The automatic semantic segmentation was based on the U-Net [26], a Convolutional Neuronal Network (CNN) architecture [27,28]. The U-Net was chosen due to its potential for biomedical image segmentation [29]. The Appendix A shows an example of appropriate segmented recorded.

To train the CNN, the training and validation set were generated from the 592 frames and masks reserved for this purpose. Out of these, 444 frames and their masks (85%) were used to compose the training set, and 148 frames and their masks (25%) the validation set. The remaining 59 frames and masks were used to create the test set and evaluate performance. Before being fed to the U-Net, frames and masks’ height and width were resized to 512 × 512 pixels, the images were normalized through min–max scaling, and one-hot encoding was applied to the multi-class masks. The open-source library TensorFlow (ver. 2.0.0 2019, Apache 2.0, Google Brain Team) was used to build and train the neural networks for automatic learning.

In the training process, two separate strategies were followed to deal with possible overfitting scenarios: data augmentation (DA) and training optimization [30]. A scheme of the training workflow can be visualized in Figure 3. DA is a technique that artificially expands the dataset by providing greater variability to the machine learning (ML) system [31,32]. Among the different DA modalities, given that medical images were employed, only the called data warping was used [33]. With this technique, new images are created transforming the data space by applying translation, rotation, and/or distortions, among others [31,34]. Upon the dataset with which the U-Net was trained, three variations of DA were used in attempt to preserve as much of the integrity of the images as possible. The image alterations include color space transformations just to the images—random changes in brightness, contrast, saturation, hue, and sharpness; geometric transformations both to the images and masks; vertical and/or horizontal axis inversion, rotations, and perspective changes, and finally, blurring through a Gaussian filter with varying kernel sizes [31].

Regarding the strategies for training optimization, in addition to the max-pooling [35] offered by the original model [29], the used methods were: dropout with a disconnection probability of 0.5, a value that is proven to provide the most optimal results [32,33,36]; learning rate decay (LRD), setting 0.1 as initial learning rate and 10^−5^ as minimum; early stopping, halting the training process, if the validation rate did not improve after eight epochs, and finally, batch normalization to regularize the data (mean = 0 and standard deviation = 1).

Up to five different configurations of the U-Net architecture were used in the training process, these can be observed in more detail in Table 1. The model which yielded the best results was when using Adam as optimizer, categorical cross-entropy loss (*L_CCE_*) combined with dice loss (*L_DLS_*) as loss function, and 24 filters in the first convolutional block.

The following equation was used to calculate the categorical cross-entropy loss ($L_{CCE})$, as described in [37]:(1)$$L_{CCE}=-\frac{1}{T}\sum_{c=1}^{C}\sum_{t=1}^{T}y_{t}^{c}\times \log\left({\hat{y}}_{t}^{c}\right)$$
where *T* is the number of samples (or pixels in this specific case), *C* the number of classes, $y_{t}^{c}$ is the target, and ${\hat{y}}_{t}^{c}\;$the model’s output.

The dice loss ($L_{DLs})$ was calculated following [38] as:(2)$$L_{DLs}=1-\frac{2\left|P\cap G\right|}{\left|P\right|+\left|G\right|}$$
where *P* stands for the predicted image (the output of the model) and *G* for ground truth or mask.

The combined loss function $L_{total}$ was calculated by means of a weighted average as:(3)$$L_{total}=\left(0.6\times L_{CCE}\right)+\left(0.4\times L_{DLs}\right)$$

All the 5 trained models used the U-Net as their framework. However, the differences between models lie in the use of DA, number of filters, and the different loss functions.

### 2.4. Feature Representation through Textures of Image

The next step was to develop a system that would allow to quantitatively determine the level of blood perfusion through the ICG intensity level in the videos. It must be considered that the reading of this fluorochrome does not yield a direct measurement, meaning that no formula directly relates ICG to the exact value of blood flow. Given that textures allow capturing the characteristics of the object’s surface in the images [39], they would enable quantifying the variations in perfusion as a function of the intensity from the ICG present in the bowel’s outermost layer. Therefore, it was decided that texture analysis of the segmented images would be the best option for the quantitative perfusion analysis.

The approach for the analysis of the textures used a combination of first- and second-order statistics. First-order statistics study individually the properties of each pixel in the image [25,39]. These descriptors are given by the mean and standard deviation of the image pixels’ intensity [39,40]. Second-order statistics study the texture through the spatial relationship between pairs of pixels [25,39,40]. To obtain these values, the spatial co-occurrence between pixel pairs is gathered through the gray-level co-occurrence matrix (GLCM) [41]. From the GLCM, up to fourteen parameters can be calculated to measure texture [25,40]. However, only seven provide essential information [39]. Hence, the used second-order statistics were: entropy, contrast, homogeneity, dissimilarity, angular second moment (ASM), energy, and correlation [25,41].

Since, from the segmented images, it was necessary to use exclusively the bowel, including the distal and proximal part, two premises were assumed for feature extraction. When determining first-order statistics, the pixels that were assigned a value of 0 (the segmentation background) were not considered. Moreover, to calculate the rest of the textures, namely the second-order statistics, all elements in the first row and column of each GLCM were eliminated. In addition, for extracting the second-order statistics, when computing the GLCM, the distance was set to 1—to calculate only the relationship between pairs of contiguous pixels—and the orientation to 0.

After successfully training the U-Net, two new videos of a segmental resection surgery—one from the Olympus^®^ and one from the Stryker^®^—were recorded, following the protocol guidelines, and selected for the quantitative analysis of blood perfusion. The videos depict the moment when, after injecting the ICG, the surgeons direct the optics of the laparoscopy tower to the area of the anastomosis, capturing the respective proximal and distal parts of the bowel (Figure 4). The videos display the area around the anastomosis and no other bowel section because, when a discoid excision or segmental resection is performed (leaving an anastomosis after removal of the affected tissue [1,9]), it is important to assess blood perfusion at that exact location [12,13]. The reason is that an adequate blood flow between the two segments of the anastomosis is crucial for the surgery to be considered successful [22].

To confirm that textures proved a reliable and useful measure for quantifying the level of blood perfusion using ICG, two experiments were conducted using the videos from each of the towers. For this purpose, 4200 frames (140 s of videos with a frame rate of 30 frames per second) of the Stryker^®^ and 3500 frames (140 s of videos with a frame rate of 25 frames per second) of the Olympus^®^ were utilized. Each of the experiments was affected separately for the two videos since, due to the NIR imaging technology used by each company, the green hue in the frames is different. However, in one version of the first experiment, the data from both videos were combined. Prior to the experiments, all frames were normalized using min–max scaling.

The first experiment was used to prove if the perfusion difference between the proximal and distal parts of each frame could be detected. For this, different supervised classifiers, specifically logistic regressors and decision trees (DT) [42], were used. The same preprocessing was followed to prepare videos. First, the proximal and distal parts of the frame were individually characterized through a 9-dimensional texture vector—the two first dimensions correspond to the first statistics, whereas the rest are associated with the second-order statistics—obtaining the vectors proximal ($\vec{p}$) and distal ($\vec{d}$), respectively. Subsequently, $\vec{p}$ and $\vec{d}$ were standardized and labeled according to the section to which they belonged (0 for distal and 1 for proximal). Finally, using an 80/20 split, the frames from each video were divided into train and test sets. During the training process, the 9-dimensional texture vectors $\vec{p}$ and $\vec{d}$ served as inputs, whereas the part they belonged to as outputs. Grid search and 5-fold cross-validation were used to find the best hyperparameters. The classification goal was to predict whether the texture vector belonged to the distal segment (class = 0) or to the proximal one (class = 1).

The second experiment was conducted to determine whether the variation in ICG intensity along the video [17] could be captured through the selected first- and second-order statistics. As in both videos, the ICG was injected with the anastomosis already present, it would not have been appropriate to consider the intestine as a continuum. Indeed, this is because in most cases, arteries in the proximal part of the anastomoses are the more affected and hence supply less blood [43]. Accordingly, to obtain a quantitative measure, each frame was characterized via a unique 9-dimensional vector that relates the two anastomosis areas. The two parts were compared by means of a Euclidean distance between the 9-dimensional texture of the proximal ($\vec{p}$) and distal ($\vec{d}$) part of the frame. As a result, each frame was characterized by a single vector $\vec{t}$:(4)$$I=||\vec{p}-\vec{d}||=\vec{t}\;\mathrm{where}\;\vec{t}=\left(t_{1}\ldots t_{9}\right)$$

The ability to contrast the ICG intensity between the two sites facilitates knowing whether one site is being less perfused than the other. Moreover, it ensures that the texture vector $\vec{t}$ is a reliable representation of the perfusion in the overall area, as both bowel segments contribute equally when computing $\vec{t}$. Afterward, to observe how the characterized frames were distributed over the video, the k-means algorithm was used for unsupervised clustering [44]. Firstly, the data were clustered in the 9-dimensional space (R^9^) using the raw texture vector $\vec{t}$. Secondly, using the unsupervised dimensionality reduction algorithms principal component analysis (PCA) [45] and t-distributed stochastic neighbor embedding (t-SNE) [46], the $\vec{t}$ vector was reduced to two (R^2^) and three (R^3^) dimensions. For each of the three vector spaces, the frames were clustered using 2, 3, 4, and 5 centroids. No correlation between the textures was detected.

## 3. Results

### 3.1. Automatic Segmentation and Performance Evaluation

For evaluating the training data of the five models, four performance metrics were utilized. Since the images suffer from class imbalance, namely the background and the predominant class, accuracy alone is not significant [47]. Hence, Matthew’s correlation coefficient (MCC) is also reported. In addition, being faced with a pixel-level classification task, two measurements designed to work with image segmentation problems (dice coefficient, DC and Jaccard index, JI) were also employed.

To demonstrate the reliability of the trained models, their performance metrics are shown in Table 2. These are complemented by four automatic segmentations of images from the test set, with the intention to provide more context and facilitate the interpretation of the outcomes. It can be observed that as improvements were included (using DA, increasing the number of filters, and employing the combined loss function), the models upgraded.

In relation to model A, it is the only one that did not learn how to segment the bowel since it classified all pixels as 0 (background class). Thus, only the accuracy is reported. However, once DA started to be implemented, the results improved as the models were able to segment the bowel. Regarding the rest of the models, adding more filters in the convolutional layers helped to improve the overall performance by almost 6%. Though, the most considerable enhancement was using the combined cost function (models C and E), as it exploits the benefits of *L_CCE_*, in the pixel level classification task, and pays attention to the morphological characteristics of the segmentations through *L_DLS_* The latter improved the performance by up to 12%. As can be seen in Table 2, the best model was E, which has an accuracy of 0.92 and a DC of 0.96.

Figure 5 shows the automatic segmentations performed on four images of the test set. Because the outputs of model A were entirely black images, their results have not been included. The rest of the segmentations were quite consistent with the expected ground truths. In the case where the proximal and distal part of the intestine must be detected, excellent outcomes were produced; those are the segmentations that ultimately do matter for the analysis of perfusion as has been previously mentioned.

### 3.2. Analysis of the Blood Perfusion Assessment through the Textures

The evaluation of the effectiveness of the textures as possible quantitative descriptors of blood perfusion, by characterizing the ICG present in the intestine, was performed separately for each of the experiments.

The first experiment verified that the perfusion difference between the proximal and distal part of each frame was indeed detected when using the nine-dimensional texture vectors $\vec{p}$ and $\vec{d}$. The classification metrics used were accuracy, F1-score, and MCC, and for the DT, the Gini impurity was also employed. Moreover, to reveal which textures were the most significant when discerning the proximal and distal parts by means of the ICG level, a study of the importance of the variables was carried out. For this purpose, to estimate the most representative features of the logistic regression, the absolute value of the weights from each variable was calculated. In the case of the DTs, the importance of the variables was obtained using the tree depth and the Gini index.

In relation to the Stryker^®^ video, the most influential features for the classification were homogeneity, the mean, and standard deviation, taking 56%, 25%, and 13% of the importance, respectively. Regarding the DTs, since the depth of the resulting tree was 1, only one feature was enough to predict which class (0 for distal and 1 for proximal) the feature vector of the frame part belonged to. In the tree, the root node was divided into two branches according to the level of homogeneity, which in turn is the most decisive texture for classification by logistic regression. When evaluating the test set, the accuracy, F1-score, and MCC were always 1, and in the case of the DTs, the Gini index for the distal part of the leaf was 0, while for the proximal part of the leaf it was 0.012.

Regarding the Olympus^®^ video, the most important features in the texture analysis highly differ from the previous. A total of 42% of the importance was given to the mean, while 30% and 22% were assigned to the contrast and dissimilarity, respectively. For the DTs, as in the previous case, the depth of the tree was 1, indicating that a single texture was sufficient to classify the two parts. In this case, the texture used was the correlation, an unexpected result since in the logistic regression only 0.5% of importance was given to it. The test set evaluation results were the same as in the previous case, with the difference that the Gini impurity was 0 for the two leaf nodes.

When training the classifiers with the frames from videos, the data are to some extent a combination of the previous. The most important features were homogeneity, the mean, the standard deviation, and dissimilarity, with each having 33%, 25%, 18%, and 13% relevance, respectively. In the DTs, again, the depth is 1; with homogeneity being the texture in charge of establishing distinction. 

By analyzing the clusters, the second experiment verified that the variation in ICG intensity throughout the video was captured using the first- and second-order statistics. Only those obtained in R^2^ space with t-SNE will be specified because the performance was similar in the three vector domains employed.

Regarding the Stryker^®^ video, when two centroids were used, the two clusters were divided by frames 1584 and 3960 of 4200. In other words, the frames before 53 s and after 132 s were classified in one group, and the intermediate ones in another. Then, it begins to stain the walls of the intestine, and after approximately 140 s, the fluorochrome begins to decay. When using three centroids, the groups became more uneven, with the frames before around 42 s belonging to one cluster, the ones in between seconds 53 and 127 to another cluster, and the remaining one in another cluster. As more centroids were used, the clusters become even more differentiated, with one cluster for frames where the ICG has not yet reached or is beginning to reach the intestine; another for the frames with very high values; and another or more clusters, depending on the number of centroids, where the rest of the frames are grouped together. Following that, in all cases, the results agreed with those expected [17], and the analysis was discontinued for more than five clusters.

The Olympus^®^ video showed the same pattern. The clusters still reflect the temporal variations reported. However, there were occasionally small groups of frames that were in a cluster to which they should not belong. These mainly corresponded to the parts in which the ICG was stabilized.

Figure 6 presents the results when using two centroids and working in R^2^ after having implemented PCA and t-SNE. Only a total of 255 frames of the Stryker^®^ video have been included for the comfort of visualization.

## 4. Discussion

The present study pretended to contribute to the gynecological surgery, for the elimination of deep intestinal infiltrating nodules, using a machine learning approach. Specifically, this study has developed a tool to quantitatively measure the level of blood perfusion in an anastomosis, for which several videos have been analyzed. This study has been divided into two axial lines.

The first line focused on the automatic segmentation of the intestine. Using a database of 651 frames and their corresponding multiclass masks, a deep learning model based on the U-Net architecture was trained to segment the bowel from the rest of the anatomical plane and to differentiate the proximal and distal parts of the anastomosis. It should be noted that the best model (model E) adequately isolates the bowel from the rest of the anatomical plane. Nevertheless, errors occur when there is no anastomosis since, when segmenting type I and III scenes, the total part tends to be misclassified as the proximal part. However, even if it must be considered, this issue is not a major problem when using the segmentations for the perfusion analysis. This could be due to in videos where there are anastomoses, the proximal and distal parts are properly segmented, and therefore the texture vectors are adequately computed. In videos where there was only the total part, the bowel can be completely reconstructed by joining the different parts segmented by the model. Therefore, it can be determined that the automatic segmentation of the bowel has been successful.

The second line centered on determining whether textures could be suitable descriptors for measuring the blood perfusion of frames based on their level of indocyanine green. To this end, the already U-Net segmented images were characterized based on the nine textures described by first- and second-order statistics. First, it was examined if the proximal and distal could be differentiated by means of the textures. The classification outcome was close to well-fit. It was later found through the feature analysis that a simple threshold in the homogeneity was sufficient to classify the different parts of the frame. Therefore, it was possible to verify that textures reflected the difference in blood perfusion between the parts of the anastomosis. Results showed how the mean is an important texture when classifying the two parts, indicating that the level of appreciable ICG present in the outermost layer of the bowel is a good descriptor of blood perfusion. Nonetheless, since other textures, such as homogeneity, dissimilarity, and correlation, play an important role in the classification, it can be reasoned that capturing other visual attributes of the bowel is crucial for a correct characterization of the pictures. Overall, considering the disparity between the importance given to textures, it is suggested that all nine statistics should be used when characterizing the frames.

Second, it was investigated whether the intensity variations within the videos could be captured when characterizing the frames via a unique texture vector (Equation (4)). The unsupervised learning algorithms k-means, PCA, and t-SNE were employed. It was found that frames belonging to the same video could indeed be clustered following the intuitions presented in [17]. The purpose of using an increasing number of centroids was to be able to observe whether, depending on the moment of the video, frames with a similar amount of ICG green staining the intestine at that time would group together. It was determined that the textures of the frames, and thus the ICG level as shown in the recordings, vary chronologically over time. Results suggest that the differences in ICG level can be gathered through the textures, demonstrating that characterizing the images based on these variables is a successful approach to quantifying the level of blood perfusion.

Comparing the current model with others to quantitatively measure ICG-perfusion, specifically FLER and Q-ICG, two major improvements should be emphasized. First, because of bowel segmentation, prior to feature extraction, textures are only computed in the bowel and not in the rest of the anatomical plane [18,19,20]. In the FLER and Q-ICG models, to produce the intensity curve from the qualitative analysis made, the intensity of all pixels in the image is calculated, thus including those regions that are not the intestine [19,20]. Although they create a single curve that is used for quantitative analysis, this is due to the FLER model, and the reference areas for ischemia and adequate perfusion must be specified [19]. In Q-ICG, the investigator must point out the areas of interest [20]. In the model presented in this article, in addition to working only on an area that is automatically segmented and differentiable between the proximal and distal part of the anastomosis, allows feature extraction to be carried out separately. By unifying the textures of both parts, it is possible to explore if there is a perfusion issue when one area is less perfused than the other. Another advantage comes from extracting a total of nine textures. In addition to having clinical relevance, since textures can capture different features of a surface, this gives the system greater versatility in capturing perfusion abnormalities that cannot be picked up by intensity itself. The combination of first- and second-degree descriptors allows ICG variations to be more accurately tracked, and the surface of the intestine to be deeply described.

### Strengths, Limitations and Future Perspectives

The main limitation of this study was the reduced number of available video recordings. As the surgeries were interrupted during the COVID-19 lockdown, there were only videos of seven surgeries when the project was developed. This has been sufficient to train an algorithm for automatic segmentation and to postulate that textures can be an adequate method to quantify blood perfusion in FGSs with bowel resection using ICG. However, it did not allow us to state certainly that these last findings were definitive. Another limitation was having to generate the multiclass masks manually, since, as is well known, this is a time-consuming and costly process. As a final limitation, an implementation challenge when training the model can be highlighted. During this process, out-of-memory errors occurred the first time when the batch of tensors was passed through the U-Net or when updating the weights through backpropagation. To overcome this problem, the batch size of the training set was reduced to 16 and in the validation to 8.

Accordingly, as future perspectives we propose: (1) to collaborate with different reference centers to be able to test the effectiveness of the automatic segmentation model, to reinforce the preliminary results on the quality of the textures as a quantitative descriptor, and to improve the protocol if this is necessary; (2) to continue recording new videos to train and improve the segmentation capacity of the model; (3) to expand the system’s application potential and use it to predict intraoperatively the success of the resection based on the level of blood perfusion in the anastomosis, and (4) once a system capable of predicting the possibility of postoperative risk was developed, to conduct a prospective study with a view to assessing the usefulness of the technique in terms of reducing postoperative complications associated with bowel resection.

## 5. Conclusions

Overall, by combining automatic bowel segmentation with texture extraction, it has been possible to quantitatively assess the level of blood perfusion of the bowel from the ICG. Critical to this was the success of CNNs, specifically U-Net, in the semantic segmentation process, and of textures in capturing the level of ICG present in the outermost layer of the bowel. For the first time, a project where deep learning and feature extraction techniques are employed is presented in the blood perfusion quantification in FGSs involving intestinal resection. Additionally, the presented protocol is a simple, reproducible, and feasible procedure that allows for properly standardizing the recording of surgical videos for subsequent analysis. Likewise, video recording and frame processing have allowed the training of deep learning algorithms and clustering analysis to validate the effectiveness of textures as blood perfusion descriptors.

This novel project on the use of deep learning and feature extraction for perfusion analysis in bowel resection surgery opens the possibility of predicting postoperative complications based on texture values and properties of sequential data in surgeries involving bowel resection and using ICG. This could help surgeons to prevent postoperative complications and act accordingly during the surgical procedure itself and considering that the final decision is always based on clinical criteria, this technique could optimize the decision. Moreover, it is feasible to use the present method in other surgeries involving intestinal anastomoses, such as colon cancer surgeries.

## Figures and Tables

**Figure 1 jpm-12-00982-f001:**
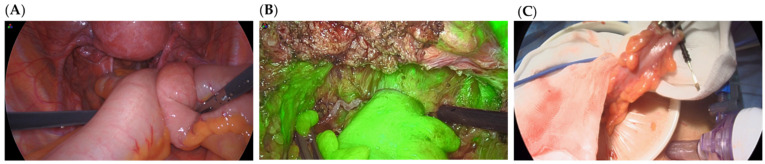
Exemplification of frame categories used to create the database. The intestinal tract in healthy conditions—in this case—or with deep infiltrating nodules (**A**). The bowel after segmental resection—in this case—or discoid excision with ICG (**B**). The proximal part of the bowel outside the peritoneal cavity after a segmental resection (**C**).

**Figure 2 jpm-12-00982-f002:**
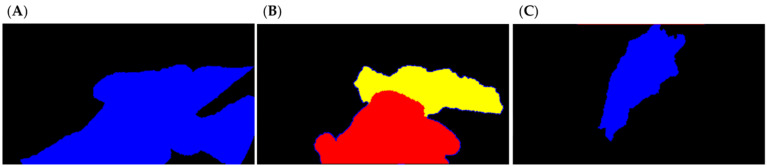
Colored multiclass masks of frames from Figure 1. Type I (**A**); type II (**B**); type III (**C**). To better distinguish, the proximal part is colored in red, the distal part is colored in yellow, and the total part is colored in blue.

**Figure 3 jpm-12-00982-f003:**
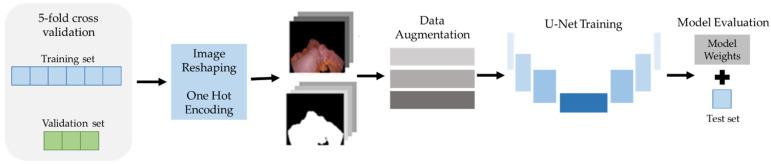
Scheme of the U-Net training and evaluation workflow considering a 5-fold cross-validation, image resampling, and a data augmentation strategy.

**Figure 4 jpm-12-00982-f004:**
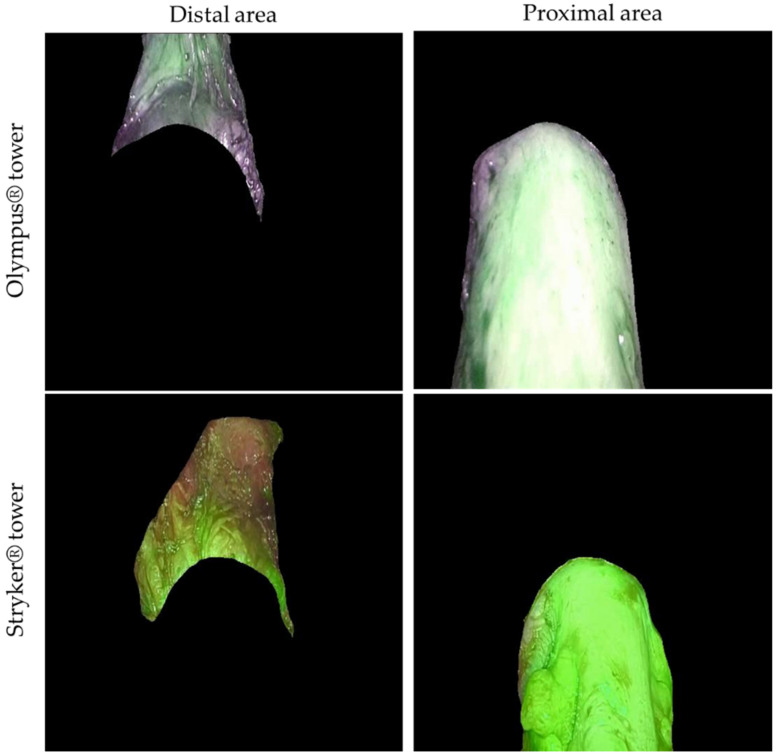
Example of the automatic segmentation, produced with the best model, of the intestine’s proximal and distal areas using the Olympus^®^ and the Stryker^®^ towers.

**Figure 5 jpm-12-00982-f005:**
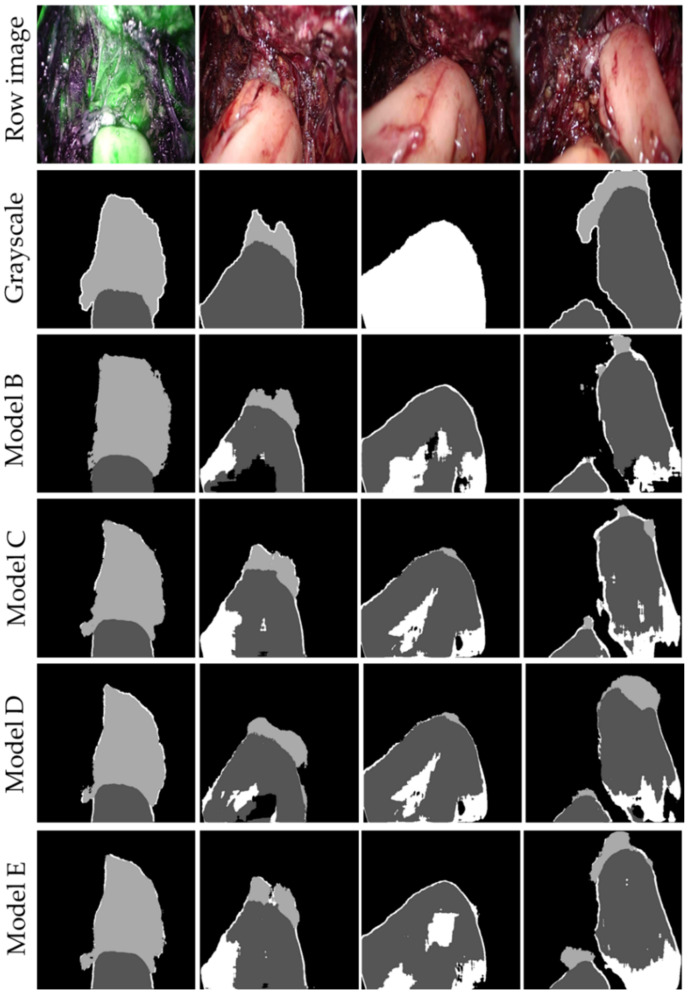
Automatic segmentations produced by models B, C, D, and E. In the top row, RGB images from the test set scaled to 512 × 512 pixels. In the next row, the ground truths show each of the 4 classes with different grey intensities. In the following rows, the masks segmented by models B, C, D, and E, respectively.

**Figure 6 jpm-12-00982-f006:**
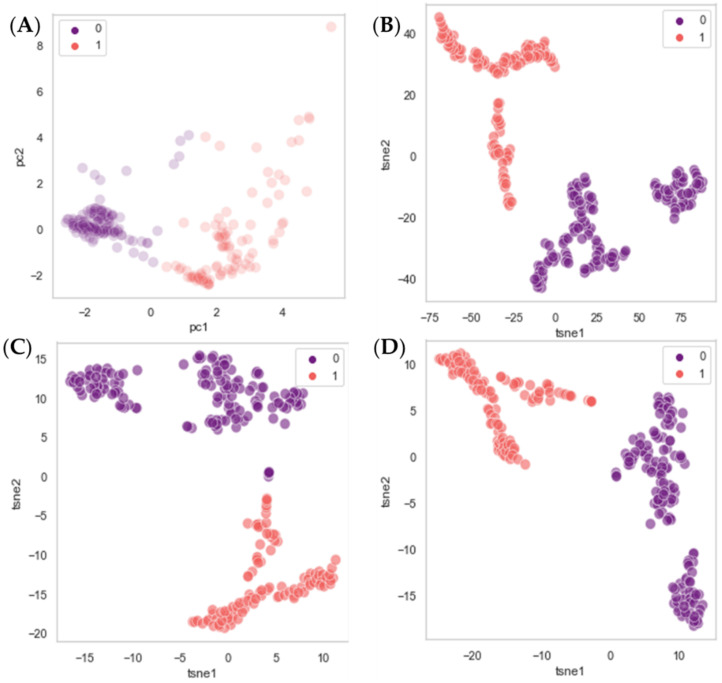
Plots obtained by clustering 255 samples from the Stryker^®^ video using k-means after dimensionality reduction using two centroids. The dimensionality reduction with PCA (**A**). The dimensionality reduction with t-SNE, using a learning rate of 200 and perplexities of 10 (**B**), 30 (**C**), and 50 (**D**). Since two centroids are being used, frames were divided into classes 0 and 1. When using PCA, the axis corresponds to the two main principal components (pc), whereas in the case of the t-SNE ones to the two main t-SNE vectors.

**Table 1 jpm-12-00982-t001:** Main differences between the trained models for the automatic segmentation task.

Model	Data Augmentation	Number of Filters	Loss Function
A	No	16	*L_CCE_*
B	Yes	16	*L_CCE_*
C	Yes	16	*L_CCE_ + L_DLS_*
D	Yes	24	*L_CCE_*
E	Yes	24	*L_CCE_ + L_DLS_*

Categorical cross-entropy loss (*L_CCE_*); dice loss (*L_DLS_*).

**Table 2 jpm-12-00982-t002:** Performance metrics of the U-Net when evaluating the images from the test set with the different trained models used to assess the training results.

Model	Accuracy	MCC	DC	JI
A	0.6715	–	–	–
B	0.7800	0.6877	0.7055	0.6875
C	0.8752	0.8183	0.8345	0.8052
D	0.8259	0.7127	0.7251	0.6903
E	0.9245	0.9478	0.9568	0.9316

Metrics to classification: accuracy and Matthew’s correlation coefficient (MCC). Metrics to segmentation: dice coefficient (DC) and Jaccard index (JI).

## Data Availability

Not applicable.

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
