# Peer review of "Deep Learning to Measure the Intensity of Indocyanine Green in Endometriosis Surgeries with Intestinal Resection"

_jpm, 2022, doi:10.3390/jpm12060982_

Round 1

Reviewer 1 Report

The usefulness of this method should be evaluated in prospective study where it can reduce postoperative bowel leakage and improve the removal of bowel endometriosis.

Author Response

The usefulness of this method should be evaluated in prospective study where it can reduce postoperative bowel leakage and improve the removal of bowel endometriosis.

Response: Thank you for taking the time to review our article. Considering your suggestion, we have implemented the discussion indicating the need to implement prospective studies (section 4.1.).

Reviewer 2 Report

In this paper, authors propose a novel method based on deep learning algorithms for quantifying the level of blood perfusion in the anastomosis. Firstly, with a deep learning algorithm based on the U-Net, models capable of automatically segmenting the intestine from the surgical videos were generated. Secondly, blood perfusion level, from the already segmented video frames, was quantified. The following review comments are recommended, and the authors are invited to explain and modify.

Comment: Novelty is confusing. A highlight is required. The main contributions of the manuscript are not clear. The main contributions of the ‎article must be very clear and would be better if summarize ‎them at the ‎end of the introduction.‎

Comment: “Binary masks of four different regions of Interest (ROI) were produced”, how to produce that?

Comment: For the preparation of binary masks, how to apply threshold on greyscale images? Didn’t that affect other organs of same intensities?

Comment: What morphological operations were applied on images?

Comment: How to optimize hyper parameters?

Comments: The design of networks and approach are from my point of view outdate, nowadays completely replaceable by a deep architecture of CNN with modern modules or a fully connected transformer. Moreover, the new approaches work very effectively and quickly even in 3D.

Comments: Model E: 0.9478, 0.9568, and 0.9316, Matthew’s correlation coefficient (MCC), dice coefficient (DC) and Jaccard index (JI). How to get these values?; and how can be these values almost equal?, authors should carefully check them.

Comment: Nothing is mentioned about the implementation challenges.

Comment: The following clinical decision support systems using Deep Learning, and medical imaging must be included to improve the quality of the paper.

·  A Lightweight Convolutional Neural Network Model for Liver Segmentation in Medical Diagnosis

· SVseg: Stacked Sparse Autoencoder-Based Patch Classification Modeling for Vertebrae Segmentation

Comment: Discuss the stability of the system in terms of complexity.

Comment: Moreover, it should be noticed that the clinical appliance has to be decided by medicals since the existing differences between the real image and the one generated by the proposed system could be substantial in the medical field.

Comment: Could you please check your references carefully? All references must be complete before the acceptance of a manuscript.

Round 2

Reviewer 2 Report

The authors have answered my questions satisfactorily.